# Pyrazole-Enriched Cationic Nanoparticles Induced Early- and Late-Stage Apoptosis in Neuroblastoma Cells at Sub-Micromolar Concentrations

**DOI:** 10.3390/ph16030393

**Published:** 2023-03-05

**Authors:** Guendalina Zuccari, Alessia Zorzoli, Danilo Marimpietri, Chiara Brullo, Silvana Alfei

**Affiliations:** 1Department of Pharmacy, Section of Chemistry and Pharmaceutical and Food Technologies, University of Genoa, Viale Cembrano, 4, 16148 Genoa, Italy; 2Cell Factory, IRCCS Istituto Giannina Gaslini, Via Gerolamo Gaslini 5, 16147 Genoa, Italy; 3Department of Pharmacy (DIFAR), Section of Medicinal Chemistry and Cosmetic Product, University of Genoa, Viale Benedetto XV, 3, 16132 Genoa, Italy

**Keywords:** neuroblastoma, cationic macromolecules, pyrazoles-based cationic nanoparticles, membrane disruptors, dose- and time-dependent cytotoxicity studies, Annexin-V apoptosis detection, 4-AAD necrosis detection, selectivity index

## Abstract

Neuroblastoma (NB) is a severe form of tumor occurring mainly in young children and originating from nerve cells found in the abdomen or next to the spine. NB needs more effective and safer treatments, as the chance of survival against the aggressive form of this disease are very small. Moreover, when current treatments are successful, they are often responsible for unpleasant health problems which compromise the future and life of surviving children. As reported, cationic macromolecules have previously been found to be active against bacteria as membrane disruptors by interacting with the negative constituents of the surface of cancer cells, analogously inducing depolarization and permeabilization, provoking lethal damage to the cytoplasmic membrane, and cause loss of cytoplasmic content and consequently, cell death. Here, aiming to develop new curative options for counteracting NB cells, pyrazole-loaded cationic nanoparticles (NPs) (BBB4-G4K and CB1H-P7 NPs), recently reported as antibacterial agents, were assayed against IMR 32 and SHSY 5Y NB cell lines. Particularly, while BBB4-G4K NPs demonstrated low cytotoxicity against both NB cell lines, CB1H-P7 NPs were remarkably cytotoxic against both IMR 32 and SHSY 5Y cells (IC_50_ = 0.43–0.54 µM), causing both early-stage (66–85%) and late-stage apoptosis (52–65%). Interestingly, in the nano-formulation of CB1H using P7 NPs, the anticancer effects of CB1H and P7 were increased by 54–57 and 2.5–4-times, respectively against IMR 32 cells, and by 53–61 and 1.3–2 times against SHSY 5Y cells. Additionally, based on the IC_50_ values, CB1H-P7 was also 1-12-fold more potent than fenretinide, an experimental retinoid derivative in a phase III clinical trial, with remarkable antineoplastic and chemopreventive properties. Collectively, due to these results and their good selectivity for cancer cells (selectivity indices = 2.8–3.3), CB1H-P7 NPs represent an excellent template material for developing new treatment options against NB.

## 1. Introduction

Neuroblastoma (NB) is a harmful embryonic neuroendocrine tumor typical in children which derives from the cells of the neural crest, from which the adrenal medulla and the ganglia of the nervous and sympathetic system physiologically develop.

Following tumors affecting the central nervous system (CNS), NB is the most common (8%) solid tumor affecting children, with an incidence of 10 cases per million each year [1]. NB is responsible for 15% of deaths ascribed to childhood tumors, and in 50% of cases, it is already metastatic and recalcitrant to chemotherapy at the time of diagnosis [1]. Generally, the site of manifestation varies depending on the age of the subject.

Particularly, in children under one year old, it is very widespread in the chest (33%), while in older infants, it is mainly diffused in the abdomen (55–75%). Although NB can appear at all ages [2], most cases have been observed in childhood, 10% of cases exceed the age of 5, and only 1.5% of patients exceed the age of 14 at the time of the diagnosis [3]. The causes of NB have not been identified so far, but the mother’s age (less than 20 years), a parent with hypertension, and the consumption in pregnancy of barbiturates and hydralazine have been recognized as risk factors [4]. Additionally, alcoholic syndrome and some morbid events, such as Hirschsprung disease and neurofibromatosis type I, have sometimes been correlated with the onset of NB. Collectively, NB includes both low-risk forms, capable of regressing spontaneously or of differentiating into benign ganglioneuroblastoma, and high-risk (HR) forms, characterized by metastatic disease and/or presence of MYCN proto-oncogene amplification [5].

Currently, treatments for HR patients include intensive and toxic chemotherapy, followed by surgical resection, myeloablation and rescue of autologous stem cells, radiotherapy, and intensive immunotherapy [6].

Usually, pharmacological treatment involves multidrug therapies consisting of cocktails of cytotoxic compounds including doxorubicin and etoposide (ETO). Unfortunately, the clinical success of the ETO-based chemotherapy is limited by its severe side effects [7,8] and chemoresistance [9].

Resistance to chemotherapeutic agents is a multifactorial phenomenon that limits the therapeutic efficacy of drugs, has a negative impact on patients’ survival and significantly affects healthcare costs [10]. To counteract the development of resistance and reduce side effects, alternative therapeutic strategies are urgently necessary. A strategic therapy should be able to exert cytotoxic effects without involving specific molecular mechanisms or enzymatic processes that tumor cells could change, thus producing a resistant phenotype. In microbiology, natural and synthetic cationic compounds, including the antimicrobial peptides (AMPs), polymers, and dendrimers, have been proven to have potent antibacterial and bactericidal effects through interacting with the negative constituents of both Gram-positive and Gram-negative bacteria’s surfaces and acting as membrane disruptors [11,12,13,14,15,16,17,18].

Interestingly, it has been found that the surface of cancer cells is very similar to that of bacteria due to the overexpression of negatively charged molecules such as phosphatidylserine, the presence of *O*-glycosylated mucines, the incidence of glycosaminoglycan side chains mainly in the form of heparin sulfate, and low levels of cholesterol. For this reason, and due to the high levels of microvilli, cancer cells attract cationic macromolecules as bacteria, and as a result are sensitive to the attack of cationic compounds [19]. Particularly, cationic compounds, simply by contact, produce irreparable damage to the cancer cell membrane, generate permeable channels, and induce mitochondrial damage. Additionally, by inducing other non-membranolytic intracellular actions, they may lead to caspase-dependent cell apoptotic death [20].

Considering the strong analogy of the cytotoxic mechanism through which cationic compounds act on tumor and bacterial cells, it is rational to think that cationic materials endowed with antibacterial/bactericidal effects could also function as antitumor agents, and that cationic polymers can offer new therapeutic opportunities in the oncological field.

In this study, two cationic copolymers prepared by us, namely P5 and P7, have been demonstrated to possess both strong bactericidal effects against multidrug-resistant clinical isolates of Gram-positive and Gram-negative species [12,21], and potent cytotoxic activity against both etoposide-sensitive (HTLA-230) and etoposide-resistant (HTLA-ER) NB cells [10].

In light of this, aiming at developing new antitumor agents to counteract NB cells, pyrazole-based nanocomposites (namely BBB4-G4K and CB1H-P7 NPs) have recently been reported to have remarkable antibacterial activity, mainly towards *Staphylococci* (BBB4-G4K NPs) [22], and broad-spectrum bactericidal effects [23]; here, we assayed them against two NB cell lines (IMR-32 and SHSY 5Y). Particularly, we carried out dose- and time-dependent cytotoxicity experiments using the pristine pyrazoles BBB4 and CB1H (Appendix A, Section S1 in Appendix A), the empty cationic NPs G4K and P7 (Appendix A) and the pyrazole-enriched nanocomposites BBB4-G4K and CB1H-P7 (Appendix A) [24,25] to assess their efficiency in inhibiting NB cells, and to assess which compound/s could be worthy of further investigations.

### 1.1. Why BBB4-G4K NPs and CB1H-P7 NPs?

The scope of a previous piece of research by us [22,23,24,25] was to develop new pyrazole-based antibacterial agents using pyrazole BBB4 (Appendix A), found active against several human diseases [24], and the free pyrazole amine (CB1) of the pyrazole hydrochloride salt (CB1H), as shown in Appendix A.

Unfortunately, both the unformulated CB1 and BBB4 showed very poor water-solubility, which would have made their administration almost impossible and limited their efficacy in any hypothetical future clinical trial. Additionally, during early antimicrobial investigations, due to its extreme insolubility in water, BBB4 did not allow us to achieve reliable results.

Therefore, we solved the drawbacks of BBB4 related to water-solubility through its nano formulation, using the highly hydrophilic and water-soluble cationic dendrimer G4K and obtaining water-soluble NPs (BBB4-G4K NPs) [24] which demonstrated remarkable and improved antibacterial properties and high values of selectivity indices (SIs) [22]. As for CB1, its poor water-solubility was solved in advance by preparing it in the form of ammonium hydrochloride salt (CB1H) [25], while its nano formulation using broad-spectrum bactericidal NPs, namely P7 [21], was carried out to improve the weak antibacterial properties observed for CB1H [23]. The obtained CB1H-P7 NPs demonstrated potent broad-spectrum bactericidal effects even higher than those of P7 [21].

### 1.2. The Rational of the Present Work

Among cationic molecules and macromolecules, natural antimicrobial peptides (NAMPs) were the pioneers that stimulated the decennial research that is currently in progress and is finalized to develop new antimicrobial agents, also effective against multidrug-resistant (MDR) bacteria, to replace traditional antibiotics. Particularly, NAMPs are short molecules that are evolutionarily well preserved and represent the main innate defense system in plants, invertebrates, and vertebrates [26,27,28].

NAMPs were used as template molecules for the synthesis of less toxic and more stable cationic polymers, co-polymers, and dendrimers with several different structures. The developed cationic macromolecules were shown to be potent antimicrobial agents.

Additionally, due to several similarities existing on the surfaces of bacteria and cancer cells, including net negative charge, NAMPs and the successively synthesized cationic materials have shown the ability to interact electrostatically, to permeabilize, to cross cell membranes and to kill both bacteria and cancer cells. In this regard, specific recognition of tumor cells is facilitated by the presence of the negatively charged phosphatidylserine (PS), which is an anionic phospholipid normally maintained on the inner leaflet of the cell membrane and externalized in malignant cells [29].

In fact, it has been demonstrated that high levels of reactive oxygen species (ROS) and hypoxia modify the tumor microenvironment and induce membrane phospholipids’ impairment [30]. Consequently, tumor cells lose the asymmetry of phospholipids’ distribution between the outer and inner layers of the cytoplasmic membrane and expose PS on its outer layer. The presence of a high number of PS on the surface of cancer cells confers to them a net negative charge, thus making them very attractive for NAMPs and the majority of synthetic antibacterial cationic compounds prepared so far, which consequently represent a new promising option for fighting tumors, NB included. As an example, Table 1 collects the most representative NAMPs also found to be active against cancer cells.

## 2. Results and Discussion

### 2.1. Cytotoxicity of BBB4-G4K on NB Cells

Dose- and time-dependent cytotoxicity experiments were performed to evaluate the effects of BBB4-G4K NPs and of CB1H-P7 NPs on IMR-32 and SHSY 5Y NB cells. We selected these cell lines as representative of NB cell lines genetically male and female, respectively. Particularly, in a first experiment, NB cells were exposed to increasing concentrations of pristine BBB4 or CB1H for 24, 48 and 72 h to assess the intrinsic cytotoxicity of the unformulated pyrazoles. Secondly, as reported in a study recently published by us [64], NB cells were treated with BBB4-G4K or CB1H-P7 NPs at the same concentrations as those of BBB4 or CB1H at the base of their drug loading and molecular weights (MW) [24,25]. Finally, cells were treated with G4K or P7 at the same concentrations as those of BBB4-G4K NPs or CB1H-P7 NPs. The aim was to assess the cytotoxicity of the empty cationic macromolecules (dendrimer G4K and copolymer P7) used to entrap BBB4 and CB1H, and to prove if the nanotechnological manipulation of BBB4 and CB1H using G4K and P7 had succeeded in improving the cytotoxic effects of the single ingredients, thus providing new promising antitumor agents. The concentrations of each sample administered to cells have been detailed in Appendix A available in Section S2 of SM. Although compounds with IC_50_ values over 30 µM were recently reported as molecules with selective cytotoxic activity against SHSY 5Y NB cells [65], to assess the actual potential of our compounds for developing new promising agents against NB cells, we take as a reference compound the N-(4-hydroxyphenyl)-retinamide or fenretinide (4-HPR).

4-HPR is a synthetic amide of all-trans-retinoic acid (ATRA), first produced in the late 1960s, which is considered a very promising antitumor agent. 4-HPR is reported to inhibit in vitro several types of tumors at 1–10 µM concentrations [66,67], including cell lines resistant to ATRA and cis-retinoic acid [68,69,70,71,72]. Particularly, a recent study reported that 4-HPR, depending on time of exposure, displayed IC_50_ = 0.68–1.93 µM towards IMR 32 cells and IC_50_ = 4.32–7.84 µM on SHSY 5Y cells [64]. In light of this, we decided to consider IC_50_ = 10 µM as the threshold over which compounds were to be considered inactive. As reported in Appendix A, the cytotoxic activity of BBB4 as a function of its concentrations was investigated using the concentration range 0.5–20 µM, which means that G4K was tested in the concentration range 0.03–1.31 µM, while BBB4-G4K NPs 0.03–1.13 µM (Appendix A). Although the concentrations used for G4K and BBB4-G4K NPs were, remarkably, <10 µM, we decided to not exceed such values, since in cytotoxicity experiments at 24 h of exposure carried out on normal human keratinocytes (HaCaT) as previously reported [22], we observed a significant reduction in cell viability already at 1 µM concentrations (G4K *p* < 0.05; BBB4-G4K *p* < 0.01). The results, expressed as mean ± standard deviation (S.D.) and obtained against IMR-32 cells, have been reported in Appendix A, while those obtained against SHSY 5Y have been reported in Appendix A.

Collectively, the cytotoxicity of all compounds towards IMR-32 NB cells did not depend either on concentration or on time of exposure. At 24 and 72 h of exposure, cell viability was over 50% at all concentrations tested for all samples, with G4K being the more cytotoxic compound, causing 37% of the cell death when administered at concentration 1.31 µM for 24 h of exposition. Because, at a concentration < 1.31 µM (1 µM), it would cause over 10% of normal cell death [22], the development of G4K as promising agent to counteract IMR-32 NB cells is not suggested.

At 48 h of exposure, a significant reduction in cell viability was observed for pyrazole BBB4 at concentrations of 7.5 and 10 µM, and for BBB4-G4K NPs at concentrations of 0.84 and 1.13 µM. However, the high fluctuation of their cytotoxic effects regardless of the increasing concentration did not support their development as effective and reliable cytotoxic agents against IMR-32 NB cells.

Towards SHSY 5Y NBs cells, the cytotoxicity of all compounds was lower than that observed towards IMR-32 cells, but it did not depend linearly on concentrations as was observed on IMR-32 cells. Additionally, at times of exposure over 24 h, significant differences in cell viability between the samples and the control were not detected, and proliferation was observed in several cases. At 24 h of exposure, a significant reduction in cell viability was observed when administering BBB4 at concentrations ≥ 12.5 µM, and when cells were treated with G4K at concentrations 0.49, 0.85 and 1.31 µM. However, cell viability was always over 50%, except when BBB4 was administered at a 20 µM concentration. In light of this result, we took into consideration BBB4 and plotted its concentration values vs. the values of cell viability, thereby obtaining the dispersion graph shown in Appendix A. Then, the equation of its linear tendency line allowed us to obtain the IC_50_ of BBB4, which was 19.9 µM, and two-fold higher the selected threshold value of 10 µM. We decided not to consider these compounds any further. 

### 2.2. Cytotoxicity of CB1H-P7 NPs on NB Cells

In cytotoxicity experiments at 24 h of exposure which were carried out by treating HaCaT cells with CB1H, CB1H-P7 and P7 NPs, we observed that NPs caused a significant reduction in cell viability only over concentrations of 1.43 µM [23]. So, in this case, the cytotoxic activity of the CB1H as a function of its concentrations was investigated using a wider concentration range (1–100 µM). Consequently, both P7 and CB1H-P7 NPs were tested in a concentration range of 0.03–2.87 µM (Appendix A).

#### 2.2.1. Cytotoxicity of CB1H-P7 NPs on IMR 32 NB Cells

The results obtained, expressed as mean ± S.D., against IMR-32 cells have been reported in Appendix A.

Curiously, P7 was more cytotoxic at 24 h of exposure than at 48 and 72 h, and at 48 h than at 72. Particularly, both at 48 and 72 h, it caused a significant reduction in cell viability at a concentration of ≥2.15 µM, and viable cells were 37% after 48 h and 41.2% after 72 h; meanwhile, at the highest concentration tested (2.87 µM), viable cells were 16.3% and 14.4%, respectively. At 24 h, P7 caused a significant reduction in cell viability already at a concentration of 0.14 µM, but its cytotoxic effects were not proportional to its concentration, and high variability in viable cells independent of the P7 concentrations was observed for doses > 0.14 µM. The less cytotoxic compound was the pristine CB1H, which caused a significant reduction in cell viability for concentrations ≥ 15 µM, both at 24, 48 and 72 h of exposure, but depending on time, an increasing percentage of cells were inhibited. Indeed, the cytotoxic effects of CB1H depended on both its concentration and time of exposure. CB1H-P7 NPs were the most effective compound in inhibiting IMR-32 NB cells, causing a significant reduction in cell viability at concentrations of ≥0.57 µM after 24 h of exposure and of ≥0.43 µM after 48 and 72 h of exposure. Similar to CB1H, the cytotoxic effects of CB1H-P7 NPs depended on both their concentration and time of exposure. Collectively, the results observable in Appendix A show that while unformulated CB1H was significantly active only at concentrations too high (≥15 µM) to be considered as a new promising compound to kill NB cells, upon its nanotechnological manipulation using P7, CB1H-P7 NPs were achieved, which were active at concentrations ≥ 0.5 µM. In this regard, CB1H-P7 NPs were 30-fold more potent than unformulated CB1H. Additionally, to better compare the activity of CB1H, P7 and CB1H-P7 NPs, and to assess the clinical feasibility of the future development of CB1H-P7 NPs as a chemotherapeutic drug against NB, we plotted the viability cells (%) vs. concentrations for all samples, obtaining the dispersion graphs reported in in Appendix A.

Then, using the equations of the linear regressions associated with the dispersion graphs obtained, we found the IC_50_ for all samples (Appendix A, Section S2, SM). The selectivity of the cytotoxic activity of the tested compounds against tumor cells (cytotoxic properties) was estimated by the selectivity indices (SI) reported in Appendix A.

The IC_50_ values reported in Appendix A evidenced that the cytotoxic effects of CB1H were dependent on the time of exposure, while those of P7 and CB1H-P7 NPs were not. Moreover, the IC_50_ values confirmed that CB1H was the less cytotoxic compound both against IMR 32 cells and also towards normal cells. Additionally, by nano-formulating CB1H with P7, we have enhanced the cytotoxic activity of both CB1H (16-30-fold more active) and P7 (about 3-fold more active) against NB cells, as well as the SI values of P7.

#### 2.2.2. Cytotoxicity of CB1H-P7 NPs on SHSY 5Y NB Cells

Appendix A available in the SM show the results, expressed as mean ± S.D., obtained against the SHSY 5Y NB cells. Unformulated CB1H caused a significant reduction in cell viability for concentrations ≥ 25 µM at 24 h, 20 µM and 15 µM at 48 and 72 h, respectively, thus evidencing that time of exposure affects its cytotoxic effects. Additionally, the cytotoxicity of CB1H for concentrations ≥ 20 µM was practically the same at 48 and 72 h, and for concentrations ≥ 50 µM, the number of viable cells was < 5% both at 24 (4%), 48 and 72 h (2%) of exposure. As for P7, at the highest concentration tested (2.87 µM), the viable cells were about 50% at 24 h of exposure, and were less than 50% at 48 h (36%) and 72 (14%) of exposure. For concentrations of P7 < 2.87 µM, the cell viability was > 50% in any situation, and in some cases, proliferation was observed. As was detected against IMR-32 cells, CB1H-P7 NPs were the most effective compound, causing a significant reduction in cell viability at concentrations of ≥0.43 µM (46% viable cells) after 24 h of exposure, of ≥0.29 µM (55% viable cells) after 48 and of ≥0.57 µM (44% viable cells) after 72 h of exposure, thus showing that the cytotoxic effects of NPs are dependent on concentration but not on time of exposure. Furthermore, viable cells were <3% (24 h) and 2% (48 and 72 h) at concentrations of ≥1.44 µM. Similar to the one plotted for IMR-32 NB cells, we also plotted the viability cells (%) vs. concentrations for all samples in this case, thus obtaining the dispersion graphs reported in Appendix A. Then, as reported above, we found the IC_50_ for all samples, which were reported in Appendix A, and estimated the SI values of all samples.

The IC_50_ values reported in Appendix A evidenced that all compounds were less active against SHSY 5Y NB cells than against IMR-32, thus having SI values lower that those observed for IMR-32 NB cells. CB1H was confirmed as the compound less active against SHSY 5Y cells and less cytotoxic towards human keratinocytes. P7, used to entrap CB1H and aiming to improve both its cytotoxic effect and its SI values, was significantly cytotoxic, displaying IC_50_ = 1.82–3.56 µM depending on time of exposure. Unfortunately, the SI < 1 in all cases did not support its development as a future new weapon to fight NB, in its empty form. On the contrary, CB1H-P7 NPs were demonstrated to be 36-44-fold (depending on time of exposure) more active than pristine CB1H, and 3-5-fold (depending on time of exposure) more cytotoxic than empty P7. Additionally, their SI values were 2.4-3.3-fold higher than those of P7, and comparable to those of CB1H, thus establishing that the nanomanipulation of CB1H using P7 led to an improvement in both the cytotoxic properties and the SI values of both CB1H and P7. Finally, CB1H-P7 NPs were over 10-fold more effective than the best-performing molecule belonging to a library of compounds whose cytotoxic activity was recently reported against the SHSY 5Y cells used in this study after 48 h of exposure [65]. As mentioned above, upon nano-formulation of CB1H using P7, the antitumor effects of both P7 and CB1H were improved. Additionally, it is unequivocal that the most important causative element of the cytotoxic activity of CB1H-P7 NPs was P7, previously reported to inhibit both etoposide-sensitive (HTLA-230; IC_50_ = 5.1 µM) and etoposide-resistant (HTLA-ER; IC_50_ = 4.0 µM) NB cells [10]. In our previous study [10], we demonstrated that P7, similar to several chemotherapeutic drugs, exerts its cytotoxic activity by increasing the production of ROS. Therefore, we would hypothesize that, also in this case, P7 contributed to the cytotoxicity of CB1H-P7 NPs, causing an increase in ROS. In this regard, it has been demonstrated that high levels of ROS in cancer cells induce membrane phospholipids’ impairment [30] and exposition of PS on the outer layer of cytoplasmic membrane. The presence of a high number of negative PS on the surface of cancer cells would have attracted cationic CB1H-P7 NPs and favored their absorption on cells surface, thus causing membrane depolarization, progressive permeabilization and irreparable damage until subsequent cell death. To obtain more reliable results, and possibly to have further confirmation of results above reported for CB1H-P7 NPs against IMR-32 and SHSY 5Y NB cells, we repeated dose- and time-dependent cytotoxicity experiments in a restricted range of concentration around the IC_50_ values previously obtained (Appendix A). The experiments were carried out using CB1H at concentrations of 5, 10, 15, 20, 25 and 50 µM. The related concentrations of P7 and CB1H-P7 NPs are available in Appendix A. The results, expressed as mean ± S.D and obtained against IMR-32 cells, are shown in Figure 1, while those obtained against SHSY 5Y cells can be found in Figure 2.

As for IMR-32 cells, except for experiments at 48 h of exposure, P7 caused a significant reduction in cell viability only at the highest concentration tested (1.43 µM), but only for 24 and 72 h of exposure was cell viability < 50%. At 24 h of exposure, CB1H and CB1H-P7 caused a statistically substantial reduction in cell viability and were significantly more cytotoxic than P7 when administered at concentrations of 10 µM (CB1H) and 0.29 µM (CB1H-P7), respectively. Although the cell death caused was similar (34–36%), CB1H-P7 was equally as cytotoxic as CB1H at a concentration 35-fold lower. The same was observed at 48 h of exposition when CB1H and CB1H-P7 caused cell death of 28–32%, both being significantly more cytotoxic than P7. Curiously, for longer periods of exposure (72 h), they caused a significant reduction in cell viability at higher concentrations (15 µM for CB1H and 0.43 µM for CB1H-P7), and viable cells were 62% (CB1H) and 51% (CB1H-P7). However, except for the highest concentrations tested (50 µM CB1H vs. 1.43 µM CB1H-P7), for concentrations ≥ 0.43 µM (24h) and ≥0.57 µM (48, 72 h), CB1H-P7 NPs were significantly more cytotoxic than CB1H at a concentration 35-fold lower. Concerning SHSY 5Y cells, at 24 h of exposure, the cytotoxicity of P7 was not proportional to its concentration, which was very low, and viable cells remained > 50% at all concentrations tested, with 84% at the highest concentration administered. When treated for 48 h, viable cells were <50% (22%) at the highest concentration used (1.43 µM), while when treated for 72 h, viable cells were <50% at a concentration of 0.43 µM, evidencing a cytotoxicity which is dependent on time of exposure. Additionally, the percentage of viable cells at the highest concentration tested was the same (22%) both after 24 h of exposure and 48. As observed against IMR-32 cells and also against SHSY 5Y cells, CB1H and CB1H-P7 were the compounds that caused the highest percentage of death. Additionally, CB1H-P7 was remarkably more potent than CB1H, causing a considerably higher reduction in cell viability, at extremely lower concentrations, and only at the highest concentration tested for both compounds, they caused similar inhibition (viable cells < 2%). Additionally, while SHSY 5Y cells were less susceptible to CB1H than IMR-32, their susceptibility to CB1H-NPs in the concentration range considered was similar to that of IMR-32. Then, following the same procedure described above, the IC_50_ values for all samples against both cell lines were computed and reported in Table 2. The dispersion graphs constructed by plotting cell viability (%) vs. the concentrations in the range of 5–50 µM (CB1H) and 0.14–1.44 µM (P7 and CB1H-P7), as well as the equations of the related linear regressions utilized to obtain the IC_50_ values, are available in SM (Appendix A).

On the basis of these new results, the cytotoxicity and consequently, the selectivity indices of CB1H were lower those that previously determined against both cell lines. On the contrary, the cytotoxicity of nanomaterials (empty P7 and the nano-formulation of CB1H with P7, CB1H-P7 NPs) was improved against both cell lines, especially that of P7. Interestingly, as opposed to all other compounds, P7 was more cytotoxic against SHSY 5Y than IMR-32. The cytotoxicity of CB1H towards both cell lines was independent of time of exposure, while that of P7 and CB1H-P7 was independent of IMR-32 cells but dependent on time of exposure to SHSY 5Y cells. Additionally, the IC_50_ values confirmed that the more effective compound against both IMR-32 (IC_50_ = 0.43–0.47 µM) and SHSY 5Y (IC_50_ = 0.47–0.54 µM) cells was the formulation developed herein. Specifically, CB1H-P7 was 54-57-fold more potent than CB1H, 2.5-4-fold more effective than P7 against IMR-32 cells, and 53.61-fold and 1.3-2-foldagainst SHSY 5Y cells, respectively. Additionally, its selectivity indices were higher in all cases than those of CB1H (1.4–2.0 times), and up to 2.9 times higher than those of P7, thus establishing that nanomanipulation of CB1H using P7 not only enhanced the antitumor activity of the two compounds, but also enhanced their selectivity for cancer cells. Additionally, for concentrations ≥ 0.43 µM, CB1H-P7 NPs were significantly more cytotoxic (*p* < 0.001) against NB cells than against normal human keratinocytes cells, thus establishing a selective antitumor effect (Figure 3).

To further assess the relevance of results obtained with CB1H-P7 NPs, we compared their cytotoxic effects with those of 4-HPR, which, as above mentioned, is considered a very promising compound to counteract several types of tumors including NB cells. To this end, dose- and time-dependent cytotoxic experiments were carried out with 4-HPR in a range of concentrations of 0.1–15 µM on IMR-32 and SHSY-5Y NB cells for 24, 48 and 72 h of exposure. Then, the IC_50_ values were computed for 4-HPR, as they were computed for CB1H-P7 NPs, for both cell lines at 24, 48 and 72 h of exposure. The results are shown in Appendix A (for dose- and time-dependent experiments) as well as in Appendix A, respectively. In Table 3, the most important data concerning the cytotoxic activity of 4-HPR have been reported and compared with those of CB1H-P7 NPs.

According to data reported in Table 3, CB1H-P7 NPs caused a significant reduction in cell viability, with statistical significance vs. control *p* < 0.001, at the same concentration of 0.43 µM, regardless the type of cells and time of exposure; this was also the case at concentrations 1.2-2.3-fold and even 1.2-11.6-fold lower than those of 4-HPR against IMR-32 and SHSY-5Y cells, respectively. Additionally, at such concentrations, the viable cells after treatment with CB1H-P7 NPs were more reduced than those after treatment with 4-HPR in the case of SHSY 5Y cells, and slightly increased in the case of IMR 32 cells. Additionally, the IC_50_ values obtained with CB1H-P7 NPs were in the range of 0.43–0.54 µM, while those of 4-HPR were in the range of 0.68–7.84 µM. The reported selectivity of 4-HPR for breast adenocarcinoma cell line MCF-7 vs. the human embryonic kidney 293 cells (HEK293) established a selectivity index value of 5 [73], thus suggesting that 4-HPR could be more selective than CB1H-P7 NPs. Additionally, being 1.6–14.5 times more potent than 4-HPR against IMR 32 and SHSY 5Y cells, CB1H-P7 NPs represent a promising nanocomposite for the future development of new agents to inhibit NB cells. As reported, the mechanism of action of fenretinide (4-HPR) is more complex than that of other retinoids, showing different pharmacological behavior [74]. Particularly, 4-HPR inhibits cell growth through apoptosis rather than differentiation, and its apoptotic effects involve generation of ROS and lipid second messengers, which are retinoic acid receptor-independent events. In this regard, to investigate if the CB1H-P7 NPs developed here could have a similar mechanism of action to 4-HPR, we assessed the effect of CB1H-P7 NPs as well as of CB1H and P7 on apoptosis and necrosis of IMR-32 and SHSY 5Y cells.

### 2.3. Effect of CB1H, P7 and CB1H-P7 NPs on Apoptosis and Necrosis of IMR-32 and SHSY 5Y Cells

Death of tumor cells can occur mainly through apoptosis (early-stage apoptosis) and necrosis (late-stage apoptosis). In fact, although it has been reported that diverse cytotoxic approaches to counteract tumors including anticancer drugs, gamma-irradiation, suicide genes or immunotherapy are mediated through induction of apoptosis in target cells [75], non-apoptotic modes of cell death such as necrosis, as well as other forms of cell death that are not classified so far, may also mediate responses to cytotoxic therapy [75]. Particularly, apoptosis and necrosis are typified by different morphological and biological characteristics. While apoptotic cells show an apoptotic body, oligonuclear DNA fragmentation, chromatin condensation, and externalization of phosphatidylserine in the undamaged cytoplasmic membrane (CM), in necrotic cells, we observe CM lysis and an apparently intact nuclei membrane [76]. Here, to know which type of cell death, apoptosis or necrosis, was induced by CB1H, P7 and CB1H-P7 NPs in IMR-32 and SHSY 5Y NB cells, we investigated how many cells made up the apoptotic or necrotic cell population using a flow cytometric method. Particularly, we examined the staining of the cells after treatment with both 7-Aminoactinomycin D (7-AAD) and fluorochrome-labeled Annexin-V (FITC-labeled Annexin-V). 7-AAD is a fluorescent cell-viability dye which is excluded from live cells with intact membranes or early-stage apoptotic cells but penetrates dead, damaged or late-stage apoptotic and necrotic cells. Once inside the cells, it binds to double-stranded DNA with high affinity by intercalating between base pairs, and it is often used as an alternative to propidium iodide (PI). On the contrary, FITC Annexin-V is a sensitive probe for identifying apoptotic cells because it binds specifically and in a calcium-dependent manner to the negatively charged phosphatidylserine (PS), which is exposed to the outer membrane in apoptotic cells in which membrane asymmetry is lost. Since with CB1H-P7 NPs, the viability of both cell lines was <50% for concentrations > 0.4–0.5 µM (Appendix A), the presence of apoptotic and/or necrotic cells was examined using P7 and CB1H-P7 NPs at a concentration 0.57 µM, which corresponds to CB1H 20 µM for 24 and 48 h of exposure. Results concerning the distribution of Annexin-V-negative (Annexin-V intensity < 10^1^) cells (living cells) and Annexin-V-positive (Annexin-V intensity > 10^1^) cells (apoptotic cells) in IMR-32, and SHSY 5Y cells untreated (CTR) and treated with CB1H, P7 and CB1H-P7 NPs, are available in Figure 4.

Particularly, Figure 4 shows that after 24 h of exposure, the distribution of Annexin-V-negative living cells and Annexin-V-positive apoptotic cells in both IMR-32 and SHSY 5Y NB cells treated with CB1H or P7 was similar to that observed in the CTR, thus establishing that both CB1H and P7 displayed insignificant apoptotic effects. Apoptosis induced by both these compounds was significantly higher than that observed in the CTR after 48 h of treatment in IMR-32 cells, while in SHSY 5Y cells, only P7 induced a significant increase in the apoptotic cell population with respect to CTR. On the contrary, when treated with CB1H-P7 NPs, the population of Annexin-V-positive apoptotic cells was remarkably higher than that observed in the CTR, in both cell lines and in all cases, in a time-dependent way.

The percentages of apoptotic and necrotic cells in both cell lines after 24 and 48 h of treatment are shown in Figure 5 (IMR-32cells) and Figure 6 (SHSY 5Y cells).

As can be seen in Figure 5 (IMR-32 cells), untreated cells (CTR) only slightly included 4-AAD and bonded with Annexin-V, thus showing low percentages of apoptosis both at 24 (9.0%) and 48 h (9.5%), and even lower percentages of necrosis (5.7% and 6.1%). Apoptosis was similar in untreated SHSY 5Y cells (9.5% and 10.9% at 24 and 48 h, respectively), while necrosis was slightly higher at 24 h (8.3%) and lower at 48 h (4.0%) (Figure 6). In IMR-32 cells treated with CB1H, we observed a time-dependent significant increase in apoptotic cells with respect to CTR (15.7% apoptosis at 24 h of exposure and 35.5% apoptosis at 48 h), while necrotic cells did not significantly increase with respect to CTR in any case (Figure 5). On the contrary, in SHSY 5Y cells, no significant increase in both apoptosis and necrosis was observed with respect to CTR, both at 24 and 48 h of exposure (Figure 6). The treatment of both IMR-32 and SHSY 5Y cells with P7 caused a significant increase in apoptosis (24.4% in IMR-32 and 15.9% in SHSY 5Y) with respect to CTR (*p* < 0.001 for IMR 32 and *p* < 0.5 for SHSY 5Y) and concerning IMR-32; this was also the case with respect to cells treated with CB1H, but only after 48 h of exposure, while the number of necrotic cells remained not significantly different from that of CTR or of cells treated with CB1H. CB1H-P7 NPs were confirmed to be the most cytotoxic compound, in which the cytotoxic effects of both apoptosis and necrosis were significantly different (*p* < 0.001) both from those in CTR and from those observed in cells treated with CB1H or P7 in all cases. Particularly, CB1H-P7 NPs caused a remarkable improvement of the apoptotic cell population both in IMR-32 (65.6–69.9%) and SHSY 5Y (74.1–85.4%) cells, depending on time of exposure. Cell death by necrosis was lower than that by apoptosis, and while IMR-32 necrotic cells were 51.7% (24 h) and 57.6% (48 h), apoptotic cells in the SHSY 5Y cell line were 61.7% at 24 h of exposure and 64.6% at 48 h. Collectively, the results displayed in Figure 5 and Figure 6 show that although the population of the cells with late-stage apoptosis/necrosis was lower than that with early-stage apoptosis, the cell death induced by CB1H-P7 was both apoptotic and necrotic, thus indicating that the treatment of NB cells with CB1H-P7 NPs led to both the suppression of cell proliferation and to membrane permeabilization, as reported in Table 1 for several NAMPs. This finding would confirm our previous assumption concerning the mechanism of action of CB1H-P7 NPs, according to which NPs, thanks to the presence of P7, increase ROS levels, thereby causing exposition of PS on cells’ surface (apoptotic cells); then, attracted by the negative charge of PS, CB1H-P7 NPs adsorb to the cytoplasmic membrane, damaging it and causing cell death by necrosis. Though we did not investigate the effects of our samples on caspases in IMR-32 and SHSY 5Y cell lines in the present study, the involvement of caspases in the apoptotic mechanism of these cells may be conceivable given that often, apoptosis in NB cells is induced by enzymes of caspase family [77,78,79,80,81].

## 3. Materials and Methods

Pyrazoles BBB4 and CB1H (Appendix A) were prepared and characterized as previously described [24,25]. Characterization results were in accordance with those previously reported, thus confirming their structure [24,25]. The fourth-generation dendrimer G4K (Appendix A), utilized for encapsulating the pyrazole derivative BBB4, was prepared, starting from the bis-hydroxymethyl propanoic acid (*bis*-HMPA) and following a multi-step procedure [24,82,83,84,85]. Its structure was confirmed by carrying out the analyses previously reported and using the same instruments previously described. Similarly, copolymer P7, used to formulate CB1H as CB1H-P7 NPs (Appendix A), was prepared and characterized as reported previously [21], achieving analytical results that confirmed its structure. The procedure to prepare the BBB4-G4K NPs (Appendix A) and CB1H-P7 NPs (Appendix A) which here are biologically evaluated as anti-tumor agents against NB cells is available in two recent papers by Alfei et al. [24,25]. Additionally, in Appendix A collect the main characterization data of BBB4-G4K NPs and CB1H-P7 NPs, respectively.

### 3.1. In Vitro Biological Experiments

#### 3.1.1. Cell Line Neuroblastoma

The human neuroblastoma cell lines IMR-32 and SH-SY5Y were grown as monolayers in complete medium (Dulbecco’s modified Eagle medium; Sigma, Milan, Italy) and supplemented with 10% *v*/*v* heat-inactivated fetal bovine serum (Gibco-Invitrogen S.r.l., Carlsbad, CA, USA) and 50 IU/mL penicillin G, 50 μg/mL streptomycin sulphate, and 2 mM L-glutamine (all reagents from Euroclone S.p.A., Milan, Italy).

#### 3.1.2. Cell Proliferation Assay

IMR-32 and SH-SY5Y were seeded in a 96-well plate from 3000 to 10,000 cells per well in 200 μL of complete medium. After 24 h, we changed the medium, and cells were treated in triplicate with CB1H, P7 or CB1H-P7 and with BBB4, G4K or BBB4-G4K, utilizing the concentrations reported in Appendix A and with 4-HPR in a concentration of range 0.1–15 µM.

All of these treatments had a final volume of 100 μL per well, and the plates were incubated for 24, 48 and 72 h at 37 °C with 5% CO_2_.

The effect on cell growth was evaluated by a fluorescence-based proliferation and cytotoxicity assay (CyQUANT^®^ Direct Cell Proliferation Assay, Thermo Fisher Scientific, Life Technologies, MB, Italy), according to the manufacturer’s instructions. Briefly, at the selected times, an equal volume of detection reagent was added to the cells in culture and incubated for 60 min at 37 °C. The fluorescence of the samples was measured using a monochromator-based M200 plate reader (Tecan, Männedorf, Switzerland) set at 480/535 nm.

#### 3.1.3. Analysis of Apoptosis and Necrosis by Annexin-V and 7-AAD Staining

SHSY 5Y and IMR-32 cells were seeded in 24-well plates (7 × 10^5^ and 7.5 × 10^5^, respectively). The day after, cells were incubated with CB1H (20 µM), CB1H-P7 (0.5748 µM) and P7 (0.5734 µM) for 24 and 48h. Cells incubated without samples were used as controls. After the required incubation time, cells were harvested and processed for apoptosis by an Annexin-V apoptosis detection kit (Invitrogen by Thermo Fisher Scientific, Waltham, MA, USA), following the experimental protocol. Briefly, they were washed twice in cold PBS and then resuspended in binding buffer (HEPES-buffered saline solution containing 2.5 mM calcium chloride) at a density of 1 × 10^5^ cells/mL. Fluorescence-labeled Annexin-V and 7-AAD (Thermo Fisher Scientific, Waltham, MA, USA) were added to these cells, and then samples were incubated for 15 min before being analyzed by flow cytometry (Gallios, Beckman Coulter Brea, CA, USA).

#### 3.1.4. Selectivity Index

The selectivity of the cytotoxic activity of the tested compounds against tumor cells (cytotoxic properties) was estimated by selectivity indices (SI). The SI values were determined using the IC_50_, as was previously reported for the same compounds against normal human keratinocytes (HaCaT) [23] and according to Equation (1).
(1)$$\mathrm{SI}=\frac{{\mathrm{IC}}_{50}\left(\mathrm{HaCaT}\right)}{{\mathrm{IC}}_{50}\left(\mathrm{IMR}32\right)}$$

### 3.2. Statistical Analysis

All experiments were performed at least three times. Each experimental condition for the biological assays was carried out in triplicate. Differential findings among the experimental groups were determined by a two-way ANOVA analysis of variance, with Bonferroni posttests, using GraphPad Prism 5 (GraphPad Software v5.0, San Diego, CA, USA). Asterisks or other indicators (see Figures captions) indicate the following *p*-value ranges: * = *p* < 0.05, ** = *p* < 0.01, *** = *p* < 0.001.

## 4. Conclusions

In this study, an excellent candidate for the future development of a new chemotherapeutic agent for use against NB was identified in a pyrazole-based nanocomposite, and its remarkable cytotoxic activity towards two NB cell lines has been reported, discussed and compared with that of 4-HPR.

Particularly, two nanocomposites (BBB4-G4K and CB1H-P7 NPs) previously found to be active against bacteria were investigated for possible antitumor effects towards IMR-32 and SHSY 5Y NB cell lines. In parallel, their components, including pyrazoles BBB4 and CB1H, respectively, as well as dendrimer G4K and copolymer P7—which were used as nanosized polymer scaffolds to formulate BBB4 and CB1H—were also assayed in the same conditions. While the BBB4-loaded G4K nanocomposite and its ingredients were all poorly active against all NB cells (thus suggesting that we do not consider these compounds any further for the development of new curative options against NB), the CB1H-P7 nanocomposite demonstrated strong cytotoxic effects on both cell lines, causing both apoptotic and necrotic cell death. Collectively, the nano-formulation of CB1H using P7 led in all cases to the enhancement of the cytotoxic effects of both CB1H and P7 (Appendix A).

Additionally, CB1H-P7 NPs demonstrated reduced cytotoxic effects on normal human keratinocytes (HaCaT) and improved SI values with respect to P7 and CB1H, considering both NB cell lines tested (Appendix A).

Notably, the IC_50_ and SI values for CB1H-P7 NPs were 0.43–0.47 µM and 2.9–3.3 against IMR-32 cells, and 0.47–0.54 µM and 2.7–2.8 against SHSY 5Y cells. Finally, we compared CB1H-P7 NPs with 4-HPR, which has been reported as an antitumor compound under a phase III clinical trial. Despite the fact that 4-HPR has shown considerable antitumor effects, mediated mainly by apoptosis against several kinds of tumors including those recalcitrant to retinoic acid treatments, the IC_50_ values obtained against NB cells for CB1H-P7 NPs were in the range of 0.43–0.54 µM, while those of 4-HPR are in the range of 0.68–7.84 µM. The results obtained in this study establish that CB1H-P7 NPs represent a promising nanocomposite for the future development of new agents to inhibit NB cells; they are 1.6-14.5-fold more potent than 4-HPR, acting both through apoptosis, as 4-HPR does, and through necrosis.

## Figures and Tables

**Figure 1 pharmaceuticals-16-00393-f001:**
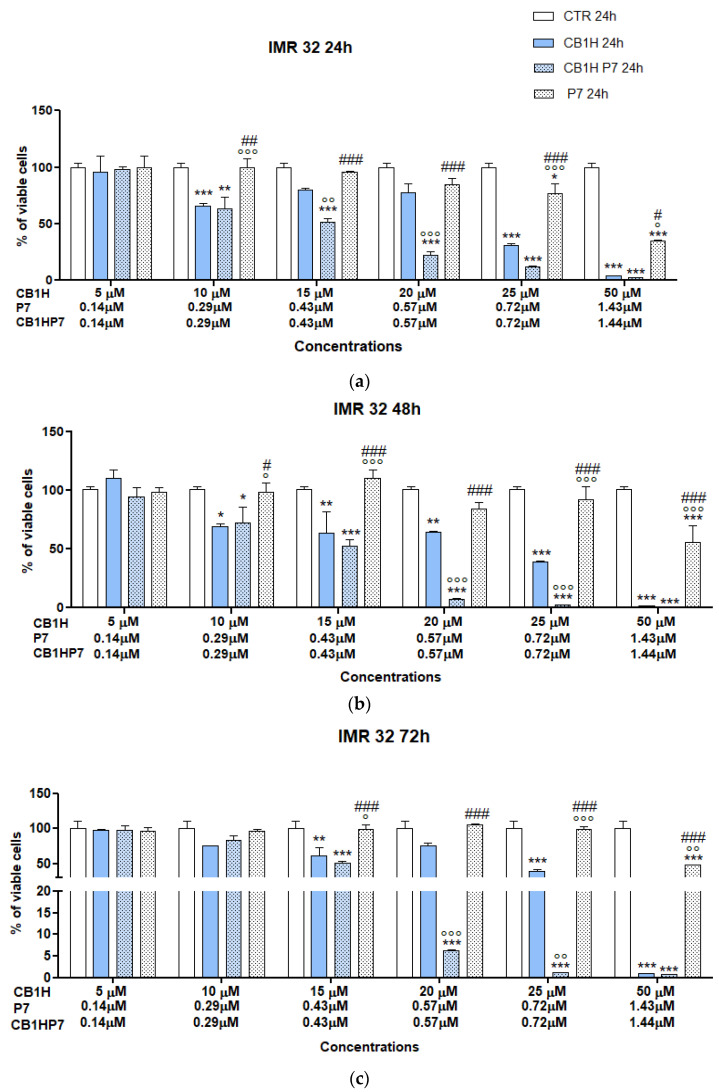
Dose- and time-dependent cytotoxicity activity of CB1H, P7 and CB1H-P7 NPs at 24 h (**a**), 48 h (**b**), and 72 h (**c**) towards IMR 32 cells. Significance refers to control (*), CB1H (°), or CB1H-P7 (#). Specifically, *p* > 0.05 ns; *p* < 0.05 * (vs. CTR), ° (vs. CB1H), # (vs. CB1H-P7); *p* < 0.01 ** (vs. CTR), °° (vs. CB1H), ## (vs. CB1H-P7); *p* < 0.001 *** (vs. CTR), °°° (vs. CB1H), ### (vs. CB1H-P7).

**Figure 2 pharmaceuticals-16-00393-f002:**
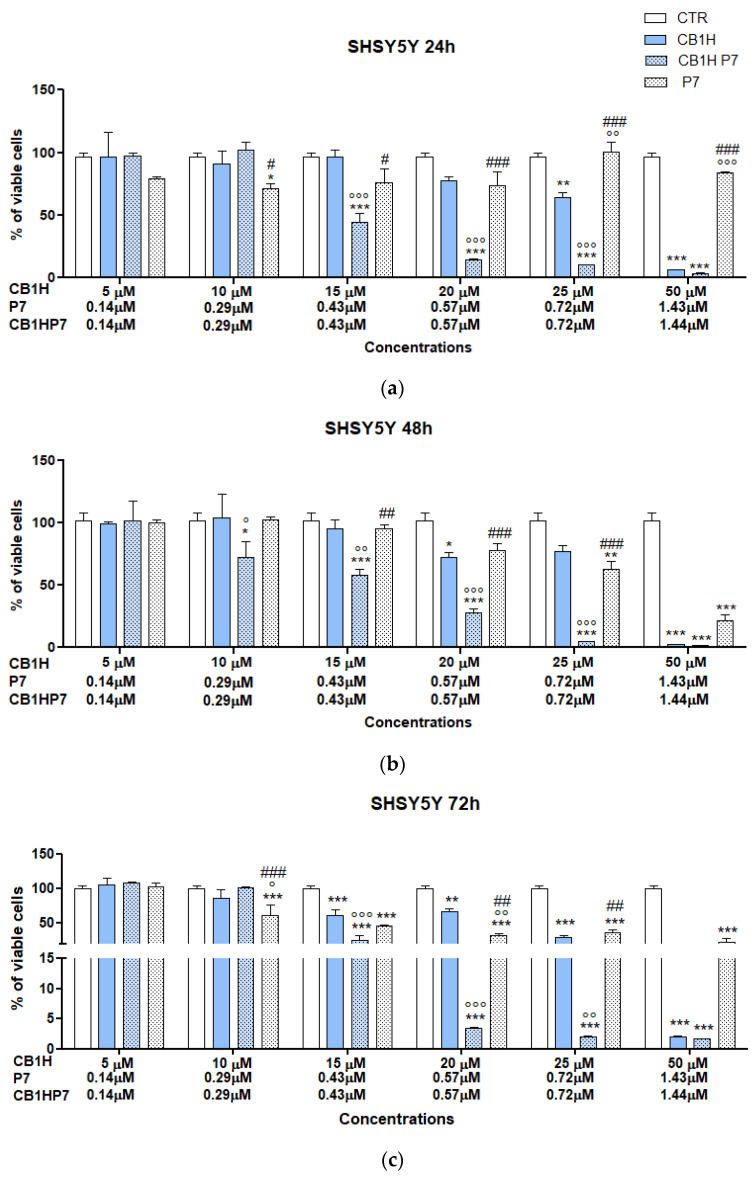
Dose- and time-dependent cytotoxicity activity of CB1H, P7 and CB1H-P7 NPs at 24 h (**a**), 48 h (**b**), and 72 h (**c**) towards SHSY 5Y cells. Significance refers to control (*), CB1H (°), or CB1H-P7 (#). Specifically, *p* > 0.05 ns; *p* < 0.05 * (vs. CTR), ° (vs. CB1H), # (vs. CB1H-P7); *p* < 0.01 ** (vs. CTR), °° (vs. CB1H), ## (vs. CB1H-P7); *p* < 0.001 *** (vs. CTR), °°° (vs. CB1H), ### (vs. CB1H-P7).

**Figure 3 pharmaceuticals-16-00393-f003:**
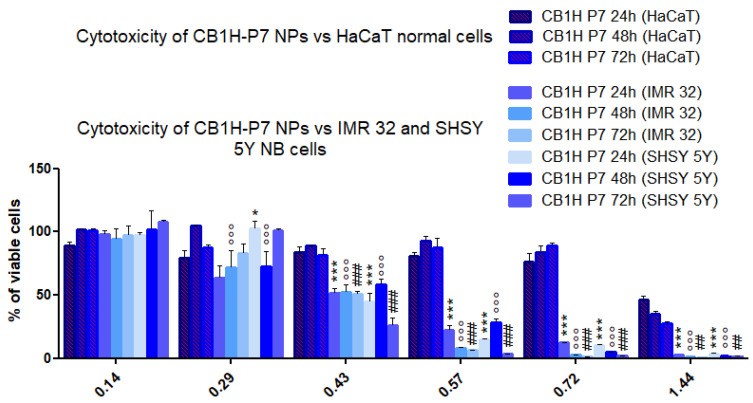
Dose- and time-dependent cytotoxicity activity of CB1H-P7 NPs at 24 h (a), 48 h (b), and 72 h (c) towards HaCaT, IMR 32 and SHSY 5Y cells. Significance refers to HaCaT cells at 24 h (*). Specifically, *p* > 0.05 ns; *p* < 0.05 * (24 h); *p* < 0.01, ## (72 h); *p* < 0.001 *** (24 h), °°° (48 h), ### (72 h).

**Figure 4 pharmaceuticals-16-00393-f004:**
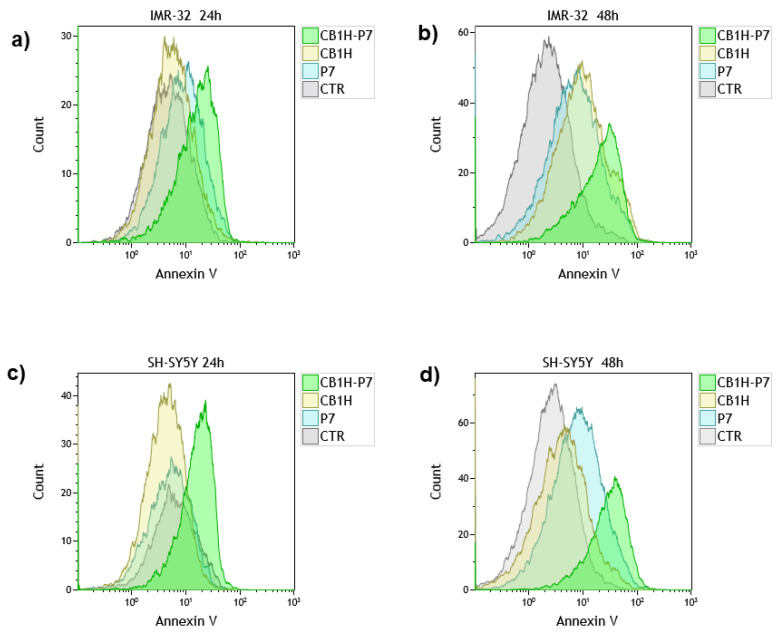
Distribution of Annexin-V-negative (Annexin-V intensity < 10^1^) cells (living cells) and Annexin-V-positive (Annexin-V intensity > 10^1^) cells (apoptotic cells) in IMR-32, and SHSY 5Y cells untreated (CTR) and treated with CB1H, P7 and CB1H-P7 NPs for 24 (**a**,**c**) and 48 h (**b**,**d**), respectively.

**Figure 5 pharmaceuticals-16-00393-f005:**
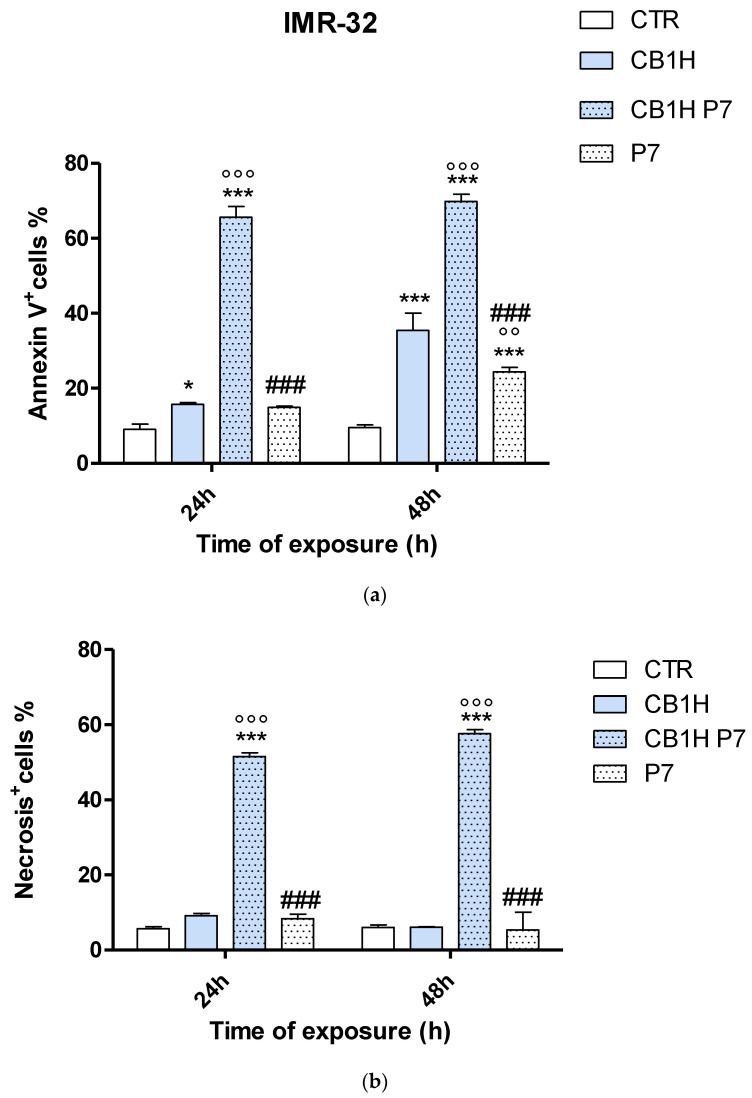
Time-dependent analysis of population of the IMR-32 cells with early-stage apoptosis (**a**) and with late-stage apoptosis/necrosis (**b**) detected by Annexin-V (**a**) and 4-AAD (**b**) staining. IMR-32 cells were grown in the presence or absence (CTR) of CB1H 20 µM, P7 0.57 µM and CB1H-P7 0.57 µM for 24 h and 48 h. Significance refers to control (*). Specifically, *p* > 0.05 ns; *p* < 0.05 * (vs. CTR); *p* < 0.01, °° (vs. CB1H); *p* < 0.001 *** (vs. CTR), °°° (vs. CB1H, ### (vs. CB1H-P7).

**Figure 6 pharmaceuticals-16-00393-f006:**
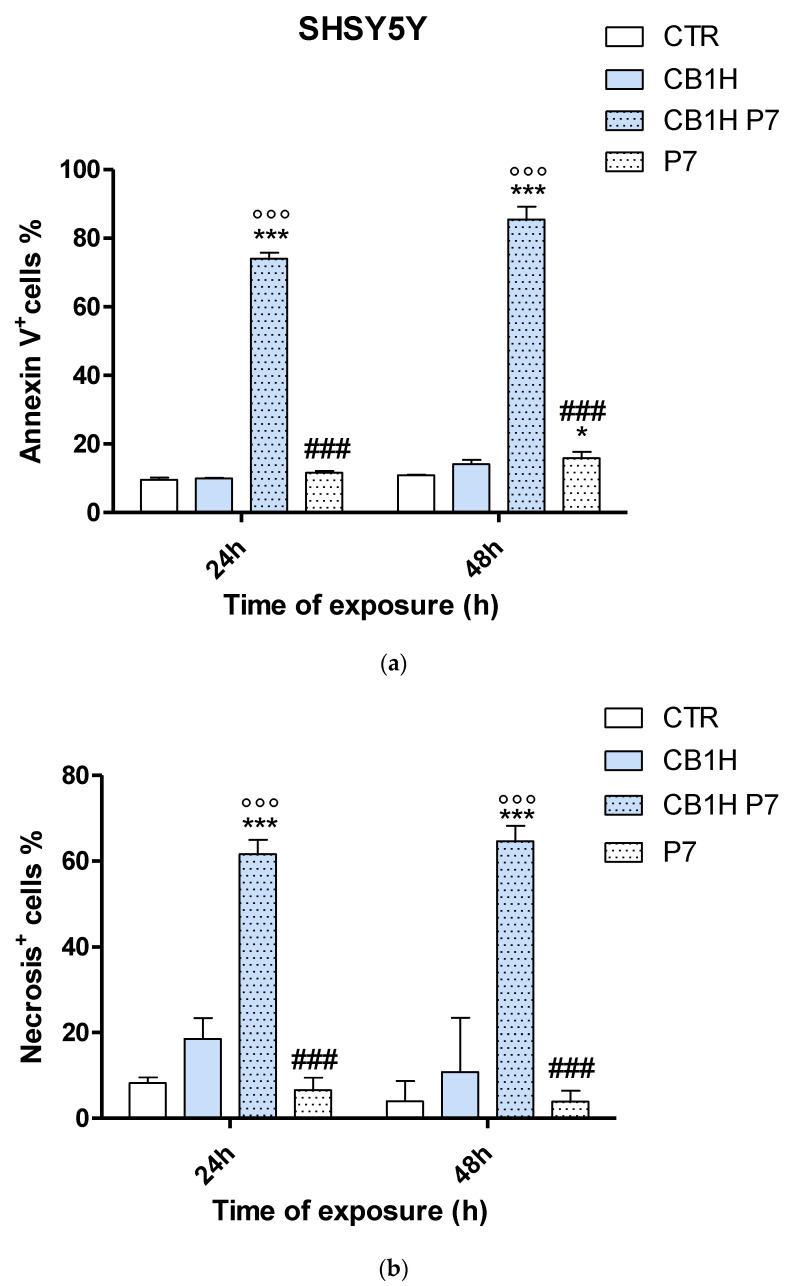
Time-dependent analysis of population of the SHSY 5Y cells with early-stage apoptosis (**a**) and with late-stage apoptosis/necrosis (**b**) detected by Annexin-V (**a**) and 4-AAD (**b**) staining. SHSY 5Y cells were grown in the presence or absence (CTR) of CB1H 20 µM, P7 0.57 µM and CB1H-P7 0.57 µM for 24 h and 48 h. Significance refers to control (*). Specifically, *p* > 0.05 ns; *p* < 0.05 * (vs. CTR); *p* < 0.001 *** (vs. CTR), °°° (vs. CB1H, ### (vs. CB1H-P7).

**Table 1 pharmaceuticals-16-00393-t001:** List of the most representative antimicrobial peptides also effective as anticancer agents.

AMP Name	Structure Class	Net Charge	Source	Tumor Target	Mechanism	Ref.
Cathelicidins LL37 hCAP18	Unknown	6	Human	HTC/STC	MP/Apoptosis	[31,32,33,34,35]
α-Defensin-1 HNP-1	Beta	3	Human	HTC/STC	Apoptosis	[36,37]
Human b-defensin-3 (hBD3)	Mixed	11	Human	HTC/STC	MP	[38]
Lactoferricin B (LfcinB)	Beta	8	Bovine	HTC/STC	MP/Apoptosis	[39,40]
Gomesin	Beta	6	Spider	STC	MP	[41]
Mastoparan-C (MP-C)	Helix	4	Venom	STC	Apoptosis	[42]
Cecropin B	Unknown	8	Silk moth	HTC/STC	MP/Apoptosis	[43,44]
Magainin 2	Helix	3	Frog	HTC/STC	MP	[45,46]
CA-MA-2	Helix	8	Hybrid	STC	MP	[47]
BuforinIIb	Unstructured	7	Frog	HTC/STC	Apoptosis	[48]
Brevenin-2R	Helix	5	Frog	STC	LDP	[49]
LFB	Unknown	4	Frog	STC	MP	[50]
Phylloseptin-PHa	Helix	2	Frog	STC	MP	[51]
Ranatuerin-2PLx	Helix	2	Frog	STC	Apoptosis	[52]
Dermaseptin-PS1	Helix	5	Frog	STC/ICD	MP	[53,54]
Dermaseptin (DPT9)	Helix	2	Phyllomedusatarsius	STC	MP	[55]
chrysophsin-1	Helix	6	Red sea bream	HTC/STC	MP	[56]
Ss-arasin	Bridge	8	Indian mud crab	STC	Uncharacterized	[57]
Turgencin A and B	Helix	3	Synoicum turgens	STC	Uncharacterized	[58]
D-K6L9	Helix	3	Engineered	STC	MP	[59]
KLA	Unknown	19	Engineered	STC	MP	[60]
LTX-315	Unknown	5	Engineered	HTC/STC	MP/ICD	[61,62]
TAT-RasGAP317-326	Unknown	8	Engineered	STC	MP	[63]

Z = pyroglutamic acid; HTC = hematological tumor cells; STC = solid tumor cells; MP = membrane permeabilization; LDP = lysosomal death pathway; ICD = immunological cell death.

**Table 2 pharmaceuticals-16-00393-t002:** IC_50_ of CB1H, P7 and CB1H-P7 NPs towards IMR-32 and SHSY 5Y cells and normal human keratinocytes (HaCaT) at 24-, 48- and 72 h exposure, computed by experiments carried out in the range of concentrations 5–50 µM (CB1H), 1–100 µM (HaCaT) and 0.14–1.44 µM (P7 and CB1H-P7). SI values were also reported.

Cells	Times (h)	CB1H (µM)	P7 NPs (µM)	CB1H-P7 NPs (µM)
IMR 32	24	25.45	1.16	0.47
	48	24.55	1.70	0.43
	72	25.05	1.63	0.46
SHSY 5Y	24	31.25	1.13	0.54
	48	31.76	1.00	0.52
	72	24.79	0.61	0.47
HaCaT *	24	57.30	2.10	1.50
	48	42.69	2.38	1.42
	72	47.63	1.71	1.33
**Selectivity Index**				
IMR 32	24	2.3	1.8	3.2
	48	1.7	1.4	3.3
	72	1.9	1.0	2.9
SHSY 5Y	24	1.8	1.9	2.8
	48	1.3	2.4	2.7
	72	1.9	2.8	2.8

* Cytotoxicity experiments were previously reported [23].

**Table 3 pharmaceuticals-16-00393-t003:** Active concentrations (statistical significance *p* < 0.001), IC_50_ values and residual viable cells (%) after treatment concerning CB1H-P7 NPs and 4-HPR.

Compound	Cells	Times	Sign. Viability Reduction (µM) ^a^	Alive Cells (%)	IC_50_ (µM)
4-HPR	IMR 32	24 h	1	28.2	1.09
		48 h	1	39.3	1.93
		72 h	0.5	40.4	0.68
	SHSY-5Y	24 h	5	45.9	7.84
		48 h	0.5	74.4	4.32
		72h	1	76.2	4.99
CB1H-P7	IMR 32	24 h	0.43	51.5	0.47
		48 h	0.43	52.6	0.43
		72 h	0.43	50.8	0.46
	SHSY-5Y	24 h	0.43	44.9	0.54
		48 h	0.43	58.0	0.52
		72 h	0.43	26.4	0.47

^a^*p* < 0.001.

## Data Availability

Data is contained within the article and Appendix A.

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
