# Peer review of "Pyrazole-Enriched Cationic Nanoparticles Induced Early- and Late-Stage Apoptosis in Neuroblastoma Cells at Sub-Micromolar Concentrations"

_pharmaceuticals, 2023, doi:10.3390/ph16030393_

Round 1

Reviewer 1 Report

This is an interesting manuscript about the nanoparticle formulation of cationic pyrazoles for the treatment of neuroblastoma. Growth in inhibition and apoptosis induction were thoroughly investigated. In my opinion, the outlined approach to a new and efficient neuroblastoma therapy is promising. I recommend acceptance after minor revision:

Some grammar and spelling errors should be corrected in a revised version of the manuscript.

Line 93: Please correct ´´Gam-positive´´.

Table 6: Please write the ´´50´´ in ´´IC50´´ in subscript.

Line 572: Replace ´´100 µl´´ by ´´100 µL´´.

Maybe the authors can discuss the mode of action of 4-HPR (positive control) and if the tested pyrazole formulations may have similar mechanisms of action.

Author Response

This is an interesting manuscript about the nanoparticle formulation of cationic pyrazoles for the treatment of neuroblastoma. Growth in inhibition and apoptosis induction were thoroughly investigated. In my opinion, the outlined approach to a new and efficient neuroblastoma therapy is promising. I recommend acceptance after minor revision:

Some grammar and spelling errors should be corrected in a revised version of the manuscript.

Line 93: Please correct ´´Gam-positive´´.

The typo has been corrected (line 93).

Table 6: Please write the ´´50´´ in ´´IC50´´ in subscript.

As asked, 50 in IC50 in Table 6 (original manuscript, now Table 3) was written in subscript (highlighted in yellow).

Line 572: Replace ´´100 µl´´ by ´´100 µL´´.

As asked, “100 µl´´ was changed in ´´100 µL´´ (line 615).

Maybe the authors can discuss the mode of action of 4-HPR (positive control) and if the tested pyrazole formulations may have similar mechanisms of action. 

We thank a lot the Reviewer for his suggestion, which appeared us very useful to complete our discussion. We have inserted the required information in Section 2.2.2 (lines 357-370 and 478-486), in Section 2.3 (lines 580-584) and in the Conclusions.

Reviewer 2 Report

This work is novel and deserve to be published in MDPI journals. But under Authors contribution you have written resources X. X.

Who is XX? 

Author Response

This work is novel and deserve to be published in MDPI journals. But under Authors contribution you have written resources X. X.

Who is XX?

We thank a lot the Reviewer for his very positive opinion and apologise for our distraction concerning the authors contribution. The mistake has been removed.

Author Response

The manuscript entitled “Pyrazole-Enriched Cationic Nanoparticles Induced Early- and Late-Stage Apoptosis in Neuroblastoma Cells at Sub-Micromolar Concentrations” describes a very interesting approach to generate novel anti-cancer formulations based on previously described antimicrobials and cationic nanoparticles and how they reduce the need to use high concentrations for future treatments. However, there are several issues that the authors must address in order to better understand the results.

We thank the Reviewer for having revised our work and for his positive general comments. Since we have noted that in his report the Reviewer required extensive editing of English language and style, we confirm that our manuscript has been revised by Prof. Deirdre Kantz, English mother tongue teacher working for University of Genoa and Pavia as linguist.

Major observations

In lines 204 – 206, you wrote “being G4K the more cytotoxic compound, which caused the 37 % of cell death, when administered at concentration 1.31 μM, for 24 hours of exposition” you need to include the information of concentrations of each compound in the figures because this is confused. I include an example of how your figures would be much more compressible.

We thank a lot the Reviewer for his suggestion which has been applied to Figure 1 and Figure 2 of the revised version of our manuscript. According to the comments and suggestions of the Academic Editor, who asked to reduce the number of Figures and Tables to a maximum of ten (Figures + Tables), several images have been moved to Supplementary Materials (SM). Anyway, the suggestion of the Reviewer has been applied also in the SM, to Figures similar to Figure 1 and 2. Please, consider Figure S4, S5, S7 and S11.

Your results suggested that P7 is the most important causative element of the cytotoxic activity shown in Tables 3 and 4, on both NB cells and normal human keratinocytes (HaCaT). You should explain more about these results in the discussion section.

The required additional discussion has been included in the main text in Section 2.2.2 (lines 357-370) and in Section 2.3 (lines 580-584).

In line 416 you wrote: “but also reduced their toxicity toward human normal keratinocytes cells”. However, this does not appear to be true, because the IC50 of CB1H-P7 NPs was also drastically reduced compared to CB1H alone in HaCaT cells. The IC50 of the mixture CB1H-P7 NPs should not have deviated too much from the IC50 of the CB1H alone, to be able to consider this statement valid.

We thank the Reviewer for his useful comment. We have changed the sentence in a more suitable one (lines 439-443).

You should include the Selectivity Index (SI) 4-HPR, reported as a compound under phase III clinical trial, and that you are using as a positive control.

The required additional information has been included in the main text (lines 473-476).

You must move the figure 15 and 16 to discussion section and only limit the concluding section to one paragraph.

As asked, Figure 15 and 16 have been moved to Supplementary Materials (SM) as Figures S25 and S26.

Minor observation

  1. In figure 1b and figure 2, the structures of CB1Ha and BBB4-G4K are observed with low resolution.

On request of Academic Editor, some Figures including Figure 1 and 2 have been removed and shifted in the Supplementary Materials.  

  1. In lines 112 to 159 about (subsections 2.1 and 2.2) must be in introduction and antecedents, these data are not the current results.

Subsections 2.1 and 2.2 have been moved in the Introduction as subsections 1.1 and 1.2 as asked. Please, see lines 158-205 (deleted part) and lines 107-155 (new insertion).

  1. In line 152 you wrote “synthtic", you should review this mistake.

The mistake has been corrected (Line 149, highlighted in yellow).

  1. In line 183 you must write in italics the phrase “in vitro”.

Done (line 229).

  1. In figures 4 and 5 you should change the commas (,) for dots (.) for example 0,5 µM for 0.5 µ (check all figures and be consistent).

As required commas were changed with dots in Figure 4 and Figure 5 (original manuscript, now Figure S4 and S5 in SM), as well as in all the Figures now present in SM where necessary.

  1. In line 452 you wrote: “g-irradiation”, you should write gamma-irradiation.

The issue has been solved (line 491).

  1. Be consistent with the IC50, 50 in subscript, or not?

All 50 in IC50 are subscripted now.

  1. Be consistent with the color of the graphs, see Fig 5 and Fig 9, Fig 11.

We make kindly note to the Reviewer that we have chosen different colour for the Figures to indicate different groups of experiments. In fact, all the Figures belonging to the first group of experiments have the same colours, those belonging to the second groups of experiments carried out in the restricted range of concentrations have the same colours, while Figure 11 (original manuscript, now Figure 3) which compares the cytotoxicity of our formulation against normal and NB cells has another colour. We ask the Reviewer to not force us at revising the colour of the images.      

  1. Place the conclusion section before the material and methods section. No point to leave before the references.

We cannot meet the request of Reviewer because in contrast with the instructions of Pharmaceuticals. In fact, we have used the template provided by the journal and we have arranged the order and type of Sections and formatted all the manuscript, accordingly.

  1. There are different line spacing in several paragraphs.

Extra spaces have been removed.

  1. A short description of the key physicochemical properties of the pyrazole-loaded nanoparticles used in this manuscript, even if the authors did the same thing that the reference [22] and name it something like “preparation (Synthesis) of the nanoparticles” (avoid auto-plagiarism).

We are not sure of having correctly understood the request of the Reviewer. Anyway, if the Reviewer asked for inserting in the work a description of the key physicochemical properties of the pyrazole-loaded nanoparticles here used as we made in the reference [22], we make kindly note him that in SM, two Tables (Table S4 and S5, revised version) reporting the required information were already present in the original version of the work, as Table S1 and S2. Please, see also lines 601-602.

Reviewer 4 Report

This manuscript explored the cytotoxicity of two cationic nanoparticles and their individual components to two neuroblastoma cell lines. They also used one health cell line as a control to study the specificity of cationic nanoparticles to tumor cells. The cationic nanoparticles were shown their capability to kill bacterial cells by disrupting cell membranes, so the authors expected a similar killing effect to human tumor cells since both cells have negative membrane charge. Generally, this study demonstrates the selective cytotoxicity of CB1H-P7 NPs to two tumor cell lines in vitro. However, I think the manuscript included many insignificant results, which makes it difficult to follow. 

Major comments:

1. I suggest reorganizing and rephrasing the results & discussion section. It's best to separate them into two sections. Also, 2.1, 2.2, and the first three paragraphs of 2.3 seem more like methods, not results. If authors want to explain their motivation and methodology, they can move these to the discussion section.

2. Since BBB4-G4K didn't show any significant treatment effect on tumor cell lines, I suggest removing these results of BBB4-G4K from the manuscript. Authors could keep them in SI and mention the results in the discussion.

3. The label of the x-axis in the most of figures are quite confusing since the x-axis only listed the concentration of one chemical but each group of bars in the figure represented the multiple results of different chemicals with different concentration. I suggest giving each condition an individual label. For example, for BBB4/BBB4-G4K/G4K, the group of 0/0/0 could be "condition 1", 0.5/0.0281/0.0328 could be "condition 2", etc. Then use condition labels on the x-axis. 

4. Annexin V can also stain necrotic cells because they have ruptured membranes. Do the authors have any control experiment to show their staining method could differentiate apoptotic and necrotic cells precisely? If so, they should include in SI.

5. Why did author repeat the dosage and time-dependent experiments with narrowed concentrations? Why didn't combine these with previous results? If the repeating tests with narrowed concentrations could show the antitumor effect of CB1H-P7 more clearly, then the previous results could be moved to SI.

6. For those figures with both very large and small bars, I suggest using a logarithmic scale of y-axis.

7. Lines 275-280, it should be moved to methods.

Author Response

This manuscript explored the cytotoxicity of two cationic nanoparticles and their individual components to two neuroblastoma cell lines. They also used one health cell line as a control to study the specificity of cationic nanoparticles to tumor cells. The cationic nanoparticles were shown their capability to kill bacterial cells by disrupting cell membranes, so the authors expected a similar killing effect to human tumor cells since both cells have negative membrane charge. Generally, this study demonstrates the selective cytotoxicity of CB1H-P7 NPs to two tumor cell lines in vitro. However, I think the manuscript included many insignificant results, which makes it difficult to follow. 

Major comments:

  1. I suggest reorganizing and rephrasing the results & discussion section. It's best to separate them into two sections. Also, 2.1, 2.2, and the first three paragraphs of 2.3 seem more like methods, not results. If authors want to explain their motivation and methodology, they can move these to the discussion section.

According to the Reviewer’s request we have extensively rephrased and reorganized the Results and Discussion Section and, while original subsections 2.1 and 2.2 have been removed and transferred to the Introduction section according to another Reviewer’s request, original subsection 2.3 has been divided into subsections 2.1 and 2.2 and the latter into subsections 2.2.1 and 2.2.2.  Then, since the instructions of the journal assert that the “Results and Discussion” Section can encompass either separate Sections or a unique Section, as we have done in several other our articles published on MDPI journals, we have chosen the second option.  

  1. Since BBB4-G4K didn't show any significant treatment effect on tumor cell lines, I suggest removing these results of BBB4-G4K from the manuscript. Authors could keep them in SI and mention the results in the discussion.

We thank the Reviewer for his useful suggestion. The results concerning the cytotoxicity of BBB4, G4K and CB4-G4K NPs (original Figure 4, 5 and 6) have been moved to SM (Figure S4, S5 and S6), while mentions to results have been maintained in the main text.

  1. The label of the x-axis in the most of figures are quite confusing since the x-axis only listed the concentration of one chemical but each group of bars in the figure represented the multiple results of different chemicals with different concentration. I suggest giving each condition an individual label. For example, for BBB4/BBB4-G4K/G4K, the group of 0/0/0 could be "condition 1", 0.5/0.0281/0.0328 could be "condition 2", etc. Then use condition labels on the x-axis. 

The Reviewer is right, and we thank him for his suggestion. According also to the suggestion of another Reviewer, we have solved the issue as observable in Figure 1 and 2 (main text revised version) and in Figure S4, S5, S7 and S11 in SM.

  1. Annexin V can also stain necrotic cells because they have ruptured membranes. Do the authors have any control experiment to show their staining method could differentiate apoptotic and necrotic cells precisely? If so, they should include in SI.

As already reported in the main text of the original manuscript, to precisely differentiate necrotic and apoptotic cells we not only evaluated the Annexin V binding but also the PI uptake (lines 487-518). Particularly, being phosphatidylserine (PS) on the inner leaflet of the intact plasma membrane of live cells, these cells do not stain with Annexin V or PI. Living and healthy cells resulted both Annexin V and PI negative. Differently, PS flips to the outer leaflet of the plasma membrane of apoptotic cells and these cells bind Annexin V on the outside but still exclude PI. Apoptotic cells were Annexin V positive and PI negative. On the contrary, cells that have undergone secondary necrosis have ruptured membranes. Annexin V binds to PS on the plasma membrane and PI is taken up and binds to the DNA. Necrotic cells were both Annexin V positive and PI positive. On these considerations we could differentiate apoptotic and necrotic cells precisely. The amounts of necrotic and apoptotic cells were counted, and results are reported in Figure 5 and Figure 6 (revised manuscript).

  1. Why did author repeat the dosage and time-dependent experiments with narrowed concentrations? Why didn't combine these with previous results? If the repeating tests with narrowed concentrations could show the antitumor effect of CB1H-P7 more clearly, then the previous results could be moved to SI.

As required, the early results have been moved to SM.

  1. For those figures with both very large and small bars, I suggest using a logarithmic scale of y-axis.

We thank the Reviewer for his suggestion, but we prefer to maintain the same scale for all bar graphs, to give uniformity to the whole work.

  1. Lines 275-280, it should be moved to methods.

The request of the Reviewer has been met (lines 637-642, highlighted in yellow).

Round 2

Reviewer 3 Report

The authors have been addressed all comments.

Reviewer 4 Report

The revision of manuscript has addressed the major issues in the previous version. I have no more comment.